# Structures, Electronic Properties and Carrier Transport Mechanisms of Si Nano-Crystalline Embedded in the Amorphous SiC Films with Various Si/C Ratios

**DOI:** 10.3390/nano11102678

**Published:** 2021-10-12

**Authors:** Dan Shan, Daoyuan Sun, Mingjun Tang, Ruihong Yang, Guangzhen Kang, Tao Tao, Yunqing Cao

**Affiliations:** 1School of Electronic and Information Engineering, Yangzhou Polytechnic Institute, Yangzhou 225127, China; shandnju@126.com (D.S.); tangmj@ypi.edu.cn (M.T.); rhyang123@126.com (R.Y.); gzkang@nju.edu.cn (G.K.); taot@ypi.edu.cn (T.T.); 2College of Physical Science and Technology, Institute of Optoelectronic Technology, Yangzhou University, Yangzhou 225009, China; sundaoyuan0808@126.com; 3Engineering Research Center of Environmental Pollutant Sensing Detection and Treatment of Jiangsu Province, Yangzhou Polytechnic Institute, Yangzhou 225127, China

**Keywords:** Si nanocrystals, electronic properties, carrier transport, temperature dependence Hall effect measurement

## Abstract

Recent investigations of fundamental electronic properties (especially the carrier transport mechanisms) of Si nanocrystal embedded in the amorphous SiC films are highly desired in order to further develop their applications in nano-electronic and optoelectronic devices. Here, Boron-doped Si nanocrystals embedded in the amorphous SiC films were prepared by thermal annealing of Boron-doped amorphous Si-rich SiC films with various Si/C ratios. Carrier transport properties in combination with microstructural characteristics were investigated via temperature dependence Hall effect measurements. It should be pointed out that Hall mobilities, carrier concentrations as well as conductivities in films were increased with Si/C ratio, which could be reached to the maximum of 7.2 cm^2^/V∙s, 4.6 × 10^19^ cm^−3^ and 87.5 S∙cm^−1^, respectively. Notably, different kinds of carrier transport behaviors, such as Mott variable-range hopping, multiple phonon hopping, percolation hopping and thermally activation conduction that play an important role in the transport process, were identified within different temperature ranges (10 K~400 K) in the films of different Si/C ratio. The changes from Mott variable-range hopping process to thermally activation conduction process with temperature were observed and discussed in detail.

## 1. Introduction

In the current studies, more and more attention has been focused on the Si nanocrystals (Si NCs) embedded in amorphous SiC (Si NCs:*a*-SiC) films due to their applications in nano-electronic and optoelectronic devices, including Si-based light-emitting diode, non-volatile memories, biosensors, and especially the Si-based solar cells [1,2,3,4,5]. It is generally accepted that SiC is beneficial to improve the performance of devices because of its lower band-gap compared with that of SiO_2_ and SiNx, which thereby contributes to the carrier transport properties [6]. Liu et al. fabricated the n-type Si NCs:*a*-SiC/p-type crystalline Si (*c*-Si) solar cell with a power conversion efficiency (PCE) reaching about 6.11% [7]. Cao et al. investigated the photovoltaic properties of Si NCs/SiC multilayers systematically. The size-controllable solar cells based on Si NCs/SiC multilayers had been prepared and showed a best PCE of 10.15% via introducing the nano-patterned Si light trapping substrates [8]. Moreover, it was firstly reported that an intense visible light emission could be observed in the Si NCs:*a*-SiC films fabricated by advanced electron-cyclotron-resonance chemical vapor deposition technique [9]. Xu et al. studied the p-i-n structures whereas the Si NCs/SiC multilayers acted as intrinsic layers, and revealed that the luminescence efficiency could be significantly improved due to their p-i-n device structures which avail the radiative recombination in intrinsic layers [10]. In order to further enhance the performance of devices based on Si NCs:*a*-SiC films, it is essential to investigate the fundamental electronic properties of Si NCs:*a*-SiC films, especially the carrier transport properties. However, the conductivity in SiC matrix, which is only about 1.9 × 10^−10^ S∙cm^−1^ for the *a*-SiC film, usually has a poor performance because of its large band gap and thus restricts the carrier transport [11,12]. It is also found that the carrier transport process is quite complicated since many factors, such as grain size, grain boundaries, crystallinity and interface states, may affect the carrier transport behaviors [13,14,15,16,17,18]. There is, thereby, an urgent need but still a significant challenge to conduct a comparable study on the carrier transport properties in Si NCs:*a*-SiC films.

In previous works, phosphorus (P)-doped Si NCs:*a*-SiC films were fabricated and both room temperature conductivity and Hall mobility were investigated [19]. For sample with low doping ratio, Hall mobility was improved with temperature, which indicated that the grain boundaries (GBs) scattering and ionized impurities scattering played a critical role in the carrier transport process. However, for samples with high doping ratios, the metalloid behavior described by a decreased mobility with temperature could be observed, implying a photon scattering dominating the carrier transport process. In the present work, Boron (B)-doped Si NCs:*a*-SiC films with fixed doping concentration and various Si/C ratios are prepared. The electronic properties association with characterization of microstructure are studied. Our results underlines that the mobilities and carrier concentrations of Si NCs:*a*-SiC films are gradually increased with the Si/C ratio, thus leading to an improvement of conductivities at room temperature. Furthermore, the temperature dependence behaviors of conductivity are also investigated. The present data demonstrate that different kinds of conduction mechanisms as Mott variable-range hopping, multiple phonon hopping, percolation hopping and thermally activation conduction contribute to the carrier transport process in B-doped Si NCs:*a*-SiC films at different temperature. The possible conduction mechanisms in films with different Si/C ratio are briefly discussed respectively.

## 2. Experiment

B-doped hydrogenated amorphous Si-rich SiC films with various Si/C ratios were fabricated by plasma enhanced chemical vapor deposition (PECVD) method using gas mixtures of pure SiH_4_, CH_4_, B_2_H_6_ and H_2_. The flow rates of SiH_4_ and B_2_H_6_ (1% diluted in H_2_) were kept at 5 SCCM (SCCM denotes standard cubic centimeter per minute) and 1.5 SCCM, respectively. The flow rate of CH_4_ was selected as 1 SCCM, 2.5 SCCM and 5 SCCM for various Si/C ratios. Here, the Si/C ratio, *R*, is defined as a gas ratio of [SiH_4_] to [CH_4_], which corresponds to 5, 2 and 1, respectively. During the preparation process, the gas-chamber pressure, substrate temperature and radio-frequency power were kept at 10 mTorr, 250 °C and 30 W, respectively. The thickness is about 200 nm for each sample. The as-deposited samples were dehydrogenated at 450 °C in N_2_ ambient for 40 min and then thermal annealed at 1000 °C for 1 h. Eventually, the Si NCs could be formed in amorphous SiC films after thermal annealing. Quartz plates were chosen as substrates for optical absorption spectrum, Raman and Hall effect measurements while p-type Si(100) wafers (thickness of 500 µm, resistivity of 1~3 Ω∙cm) were adapted to transmission electron microscopy (TEM).

Raman scattering spectra were measured by a Jobin Yvon Horiba HR800 spectrometer (HORIBA Jobin Yvon Co., Paris, France) where an Ar+ laser with a wavelength of 514 nm was used as excitation light source. The high-resolution TEM images were observed by a TECNAI G2F20 FEI high-resolution transmission electron microscopy (FEI Co., Hillsboro, TX, USA). The optical band-gap, which can be deduced by Tauc plots based on an optical absorption spectrum, was measured using a Shimadzu UV-3600 spectrophotometer (Shimadzu Co., Kyoto, Japan). Temperature dependences of conductivities were measured by Hall effect measurements at the temperature from 10 K to 400 K with the steps of 20 K below room temperature and 10 K above room temperature, which was carried out by a LakeShore 8400 HMS (LakeShore Co., Lorain, OH, USA) using van der Pauw (VDP) method.

## 3. Results and Discussion

### 3.1. Nanostructure

Raman spectra of B-doped Si NCs:*a*-SiC films with various Si/C ratios are shown in Figure 1. As a reference, the Raman spectrum of as-deposited sample is also plotted. Abroad peak locating at 480 cm^−1^, which is relevant to the transverse optical (TO) phonon mode of amorphous Si-Si phase, can be observed in the as-deposited sample. After thermal annealing, the annealed samples exhibit a sharp and strong Raman peak around 520 cm^−1^, which indicates a structural transformation from amorphous Si to crystalline Si. It can be found that the position of Raman peak is slightly downshifted with respect to the peak position of mono-crystalline Si (520 cm^−1^) by about 3 cm^−1^ for the sample with *R* = 1 and 1 cm^−1^ for the sample with *R* = 5. The red shift compared with a Raman peak from mono-crystalline Si reveals the confinement of optical phonons due to the small grain size in films [20]. According to the empirical formula: $D_{R}=2\pi \sqrt{B/\Delta \omega }$, where $D_{R}$ is the mean size of Si NC in diameter, *B* is 2.24 cm^−1^ for Si and Δω is the Raman shift of crystalline peak from mono-crystalline peak located at 520 cm^−1^, respectively [21], the Raman peaks shift to lower wave numbers by values of Δω = 1~3 cm^−1^ may be caused by Si NCs sizes of around 5.4~9.4 nm, which indicates the mean size of Si NCs are increased with the Si/C ratio in B-doped Si NCs:*a*-SiC films. Notably, it can be also characterized that the Raman intensity is gradually enhanced with Si/C ratio. In order to estimate the crystallinity of annealed samples, crystalline volume fraction (*X_c_*) is calculated according to the formula: $X_{c}=I_{c}/\left(I_{c}+I_{a}\right)$, where $I_{a}$ and $I_{c}$ are the amorphous and crystalline part of the integrated Raman scattering intensity, respectively [22]. From the results we have obtained that *X_c_* is about 56% for the B-doped Si NCs:*a*-SiC film with *R* = 1, and is gradually increased to 90% when *R* goes up to 5. On the basis of our findings, it can be concluded that higher Si content in amorphous SiC films are helpful for the formation of Si NCs after thermal annealing.

In order to gain more insight into the microstructures of B-doped Si NCs:*a*-SiC films with various Si/C ratios, TEM observations were investigated as shown in Figure 2. It can be clearly identified that the Si NCs have been taken shape in *a*-SiC films. The thickness of annealed sample is about 200 nm, which is in good agreement with the predesigned value estimated by the deposition rate. The average grain size of Si NCs, which is about 5 nm for the sample with *R* = 1, is increased with *R* and reached to larger than 10 nm for the sample with *R* = 5. It can be found that a large number of Si NCs are spread over the sample with *R* = 5, indicating a good crystallization in the film. Meanwhile, compared with the sample with *R* = 1, it is apparent that the samples with *R* = 2 and 5 have relatively high crystallinity due to the higher content of Si in the films, which are consistent with the observation from Raman results.

The optical bandgap $E_{g}$, of which the value can be deduced from the Tauc plot, is generally used to analyze the light absorption in amorphous and nano-crystalline semiconductor films [23,24]. Figure 3 plots the Tauc’s plots of ${\left(\alpha h\upsilon \right)}^{1/2}$ versus photon energy hυ for B-doped Si NCs:*a*-SiC films with various Si/C ratios. It can be found that the optical band-gaps are 2.1 eV and 2.2 eV for the samples with *R* = 1 and 2, respectively. Given that the optical band-gap of amorphous SiC film is only 1.8 eV, the higher value of $E_{g}$ for the present samples after annealing should be ascribed to the entrance of N or O into the annealed films during thermal annealing [25]. Moreover, as the Si/C ratio is further increased, the value of $E_{g}$ is obviously raised and reached to 2.8 eV for the sample with *R* = 5, which can be explained as the contribution from the GBs in the annealed films [26]. In our previous works, we observed a large amount of disordered GBs regions existing in the annealed films, which had a higher optical gap compared with the amorphous and nano-crystalline Si and might play an important role in the overall optical gap of the films [25,27]. Therefore, the increase of optical band gap with Si/C ratio can be ascribed to the appearance of large numbers of GBs in the films, likely due to the enhanced crystallization as we mentioned before.

### 3.2. Room Temperature Electronic Properties

In order to achieve better understandings of the electronic properties in B-doped Si NCs:*a*-SiC films with various Si/C ratios, room temperature Hall mobility, carrier concentration as well as conductivity were measured as indicated in Table 1. It can be clearly seen that Hall mobility and carrier concentration are gradually promoted from 1.7 cm^2^/V∙s and 8.7 × 10^16^ cm^−3^ for the sample with *R* = 1 to 7.2 cm^2^/V∙s and 4.6 × 10^19^ cm^−3^ as the *R* up to 5, which is consistent with that of doped Si NCs:*a*-SiC films reported previously [12,19]. Usually, the carrier mobility mainly depends on the defect states associated with the dislocations in Si NCs films, which significantly reduce the carrier lifetime and deteriorate the electrical transport property [26,28,29,30]. Therefore, the Hall mobility raised with Si/C ratio in B-doped Si NCs:*a*-SiC films likely owing to a gradually improving crystallization of the present samples with increasing Si/C ratio. Furthermore, more and more B impurities enter into the cores of Si NCs substitutionally to provide free carriers because of the promotion of crystallization in the present samples as increasing Si/C ratio, consequently leading to a significant increase in carrier concentration. Thus, the conductivity generally determined by the Hall mobility and carrier concentration is improved obviously as increasing Si/C ratio and reached to a maximum value of 87.5 S∙cm^−1^ for the sample with *R* = 5, which is consistent with the observation from our previous works [19].

### 3.3. Temperature-Dependent Conductivity

To further explore the carrier transport mechanisms of B-doped Si NCs:*a*-SiC films with various Si/C ratios, temperature-dependent conductivities were measured as shown in Figure 4. We note that all the samples exhibit a linear relationship of the lnσ versus $T^{-1}$ plot at the temperature from 300 K to 400 K, which is well fitted by the Arrhenius plots $\sigma =\sigma_{0}exp\left(-E_{a}/k_{B}T\right)$, where $\sigma_{0}$ is the conductivity prefactor and $k_{B}$ is Boltzmann’s constant. Our findings lead us to indicate that the carrier transport in B-doped Si NCs:*a*-SiC films above room temperature is dominated by the thermal activation conduction in the extended states [31]. Meanwhile, the value of activation energy $E_{a}$ can be obtained from the slope of lnσ versus $T^{-1}$ curve, which indicates the energy difference between the top of valence band and the Fermi level in p-type semiconductor [32]. As shown by dashed line in Figure 4, the activation energy $E_{a}$ is only 110 meV for the B-doped Si NCs:*a*-SiC film with *R* = 1 and further decreased with *R* increasing, which suggests that the Fermi level should be near and then shifts closer to the top of valence band. It is well known that the crystallization of B-doped Si NCs:*a*-SiC films can be improved by increasing Si/C ratio, leading to an improvement of B doping efficiency which has been discussed above. Consequently, the decrease of $E_{a}$ can be attributed to the increase of B dopants occupying the inner sites of Si NCs that gradually shifts the Fermi level to the top of the valence band in the present samples.

The temperature-dependent conductivities of B-doped Si NCs:*a*-SiC films with *R* = 1 and 5 were also measured below room temperature (10 K~300 K). It can be found that the conductivity data for both films are not well fitted by the Arrhenius relationship, implying different mechanisms dominating the carrier transport process below room temperature. In order to extract information about low temperature transport behaviors of B-doped Si NCs:*a*-SiC films, the reduced activation energy, $w\left(T\right)=d\left(ln\sigma \right)/d\left(lnT\right)$, is introduced and plotted against T on a log-log scale below room temperature [33,34]. An interesting finding is that different $w\left(T\right)\sim T$ relationships are identified as shown in Figure 5. When the temperature is below 50 K, the slope ≈0.25 of the $w\left(T\right)\sim T$ plots is achieved in both films, which implies the Mott variable-range hopping (Mott-VRH) conduction mechanism according to the formula: $\sigma =\sigma_{0}exp[{\left(-T_{0}/T\right)}^{1/4}]$ [35]. It is reasonable since the charge carriers hopping between the neighboring Si NCs are impeded due to the freeze out of acoustic phonons as the temperature goes down to 50 K. At the lower temperature, the charge transport is proposed as the tunneling process of charge carriers between localized states according to Mott’s model [35]. The Mott-VRH conduction had been usually reported in un-doped and P-doped Si and Ge NCs films at low temperature as well [27,36,37,38].

However, once the temperature is above 50 K, the temperature dependence behaviors of conductivity change again and are no longer described by the Mott-VRH process. For the B-doped Si NCs:*a*-SiC film with *R* = 1, the reduced activation energy $w\left(T\right)$ is found to a span of zero slope within the temperature region from 50 K to 240 K, which refers to a power–law relationship between conductivity and temperature ($\sigma =T^{\gamma }$) as shown in Figure 5. This behavior of conductivity represents the multiple phonon hopping (MPH) conduction mechanism owing to the weak carrier-phonon interactions [17,39]. In this temperature region, electrons localized weakly in the defect states can be preferentially coupled to the phonons whose wavelengths are close to their localization length [40]. Hence, the localized electrons can hop between the deep localized states originated from the defect states via the multiple phonon hopping process. Identical behaviors were also found in our previous works of the P-doped Si NCs/SiO_2_ multilayers, where the grain size of Si NCs was less than 7 nm and the crystalline volume fraction was below 70% [37]. Combined with the previous works, our findings lead us to conclude that the transport process at relatively low temperature region (50 K~240 K) is mainly dominated by the MPH conduction mechanism in the B-doped Si NCs:*a*-SiC film with *R* = 1, where the grain size is relatively smaller (about 5 nm) and the crystallinity is relatively lower ($X_{c}$ < 60%).

Nevertheless, for the B-doped Si NCs:*a*-SiC film with *R* = 5, the slope ≈ 0 of the $w\left(T\right)\sim T$ plot is no longer shown for the temperature dependence behavior of conductivity at the temperature above 50 K, but is replaced by an irregular curve. Instead of the reduced activation energy, it is useful to plot lnσ as a function of $T^{-1/\alpha }$ for the present sample and obtain a proper value of α which straightens out experimental curves. As shown in Figure 6 (insert), the lnσ ∝ $T^{-1/4}$ (α=4) behavior defined as the Mott-VRH process could be found at the temperature from 10 K to 50 K, which is consistent with the above discussion. Once the temperature is in the region from 90 K to 270 K, the value of α is changed to 2 as plotted in Figure 6. Here, the lnσ ∝ $T^{-1/2}$ behavior is expressed by $\sigma =\sigma_{0}exp[{\left(-T_{0}/T\right)}^{1/2}]$, which is usually explained by the percolation hopping model [41,42]. In this model, the semiconductor nanocrystals are dispersed in insulating matrices where they are separated from each other by a finite barrier. The thermally activated charges can transport from one nanocrystal to another by tunneling the barrier [43]. The percolation-hopping model was often reported in the similar works of Ge NCs, where the Ge NCs films had large grain sizes (>10 nm) and high crystallinities (*X_c_* > 90%). It was found that nanocrystalline material contained a large number of nanocrystals joined together by grain boundaries with a lot of defects, which could trap the carriers and then formed the potential barriers. Thus, the transport process was considered to be a tunneling of free charges between the neighboring Ge NCs at the relatively low temperature [27,41]. Consequently, the percolation hopping model can be suitable for explaining the carrier transport mechanism in the present sample with *R* = 5 because Si grains with large size in such quality crystallization film are separated by the grain boundaries as shown in the TEM images.

## 4. Conclusions

In summary, B-doped Si NCs:*a*-SiC films with various Si/C ratios were fabricated by PECVD method. The microstructures together with electric properties were investigated. It is found that both crystallinity and average grain size of B-doped Si NCs:*a*-SiC films are increased with the Si/C ratio. Furthermore, Hall mobility and carrier concentration are also found to be increased with Si/C ratio and reached to 7.2 cm^2^/V∙s and 4.6 × 10^19^ cm^−3^ as the Si/C ratio rising to 5, which is mainly due to the improvement of crystallinity as well as the improved B doping efficiency in films. Carrier transport properties in B-doped Si NCs:*a*-SiC films were studied via the temperature dependence Hall effect measurements at the temperature from 10 K to 400 K. It is interesting to find that different kinds of transport behaviors are presented in different temperature regions. We learn that the carrier transport process is dominated by the thermal activation conduction mechanism above 300 K while governed by the Mott-VRH conduction mechanism below 50 K in the B-doped Si NCs:*a*-SiC films. However, within the temperature from 50 K to 300 K, there are two different conductive behaviors observed in the films with different Si/C ratios. MPH conduction mechanism play an important role in the transport process of B-doped Si NCs:*a*-SiC films with *R* = 1 where the grain size is relatively smaller as well as the crystallinity is relatively lower, and the percolation hopping model dominates the transport process in the B-doped Si NCs:*a*-SiC film with *R* = 5, in which the clear grain boundaries can be identified.

## Figures and Tables

**Figure 1 nanomaterials-11-02678-f001:**
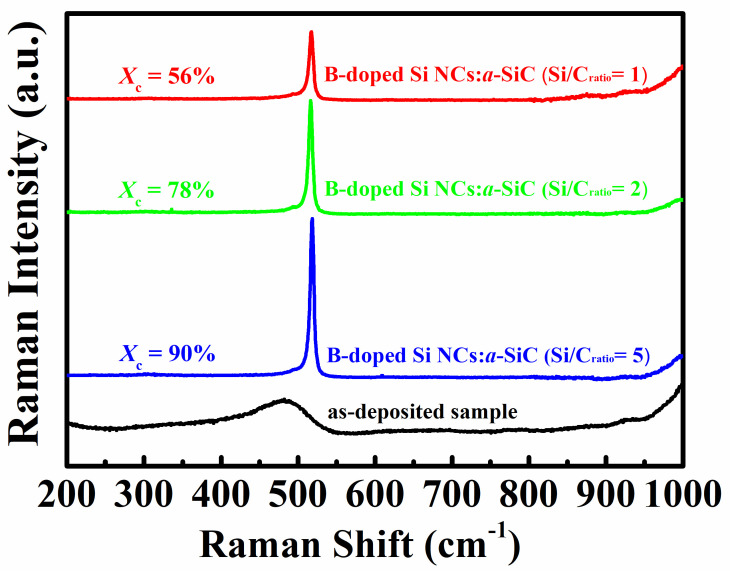
Raman spectra of B-doped Si NCs:*a*-SiC films with various Si/C ratios.

**Figure 2 nanomaterials-11-02678-f002:**
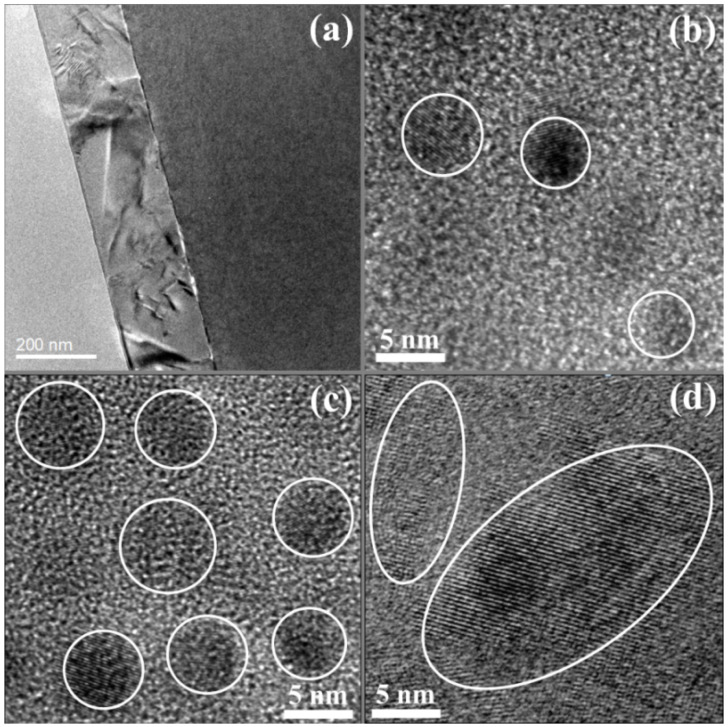
Transmission electron microscopy (TEM) images of B-doped Si NCs:*a*-SiC films with various Si/C ratios, including (**a**) cross-sectional TEM image of annealed sample; and (**b**) *R* = 1; and (**c**) *R* = 2; and (**d**) *R* = 5.

**Figure 3 nanomaterials-11-02678-f003:**
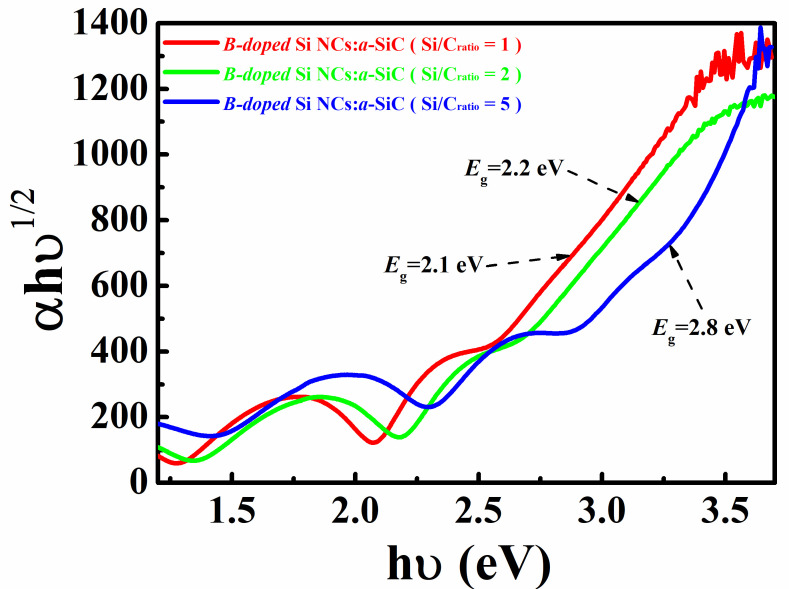
Tauc’s plots of (*αhυ*)^1⁄2^ versus photon energy *hυ* for B-doped Si NCs:*a*-SiC films with various Si/C ratios.

**Figure 4 nanomaterials-11-02678-f004:**
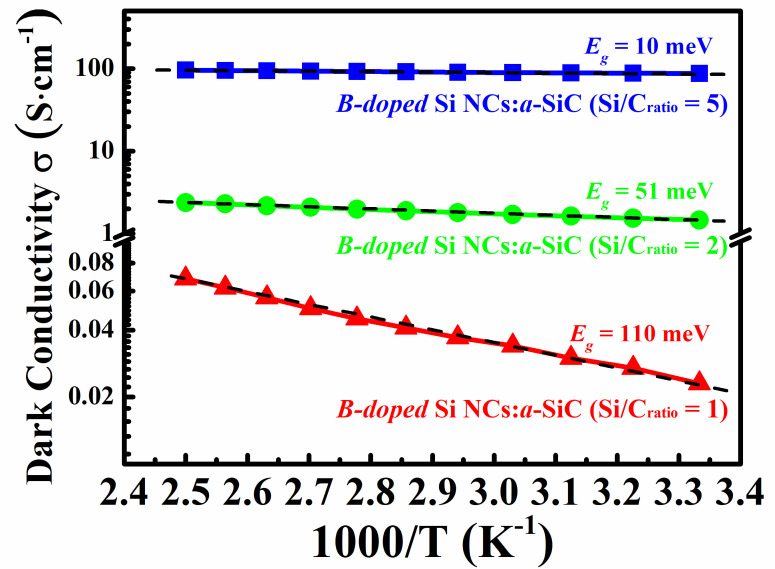
Temperature-dependent conductivities of B-doped Si NCs:*a*-SiC films above room temperature.

**Figure 5 nanomaterials-11-02678-f005:**
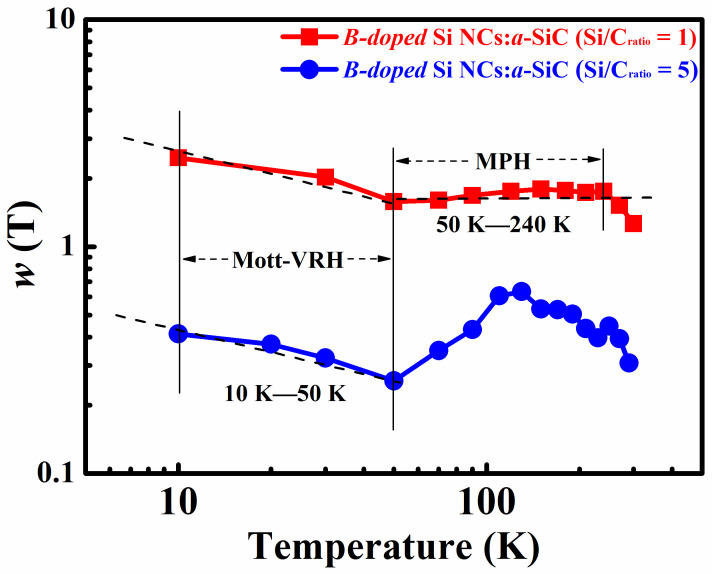
Temperature-dependent conductivities of B-doped Si NCs:*a*-SiC films with *R* = 1 and 5 below room temperature (10 K~300 K).

**Figure 6 nanomaterials-11-02678-f006:**
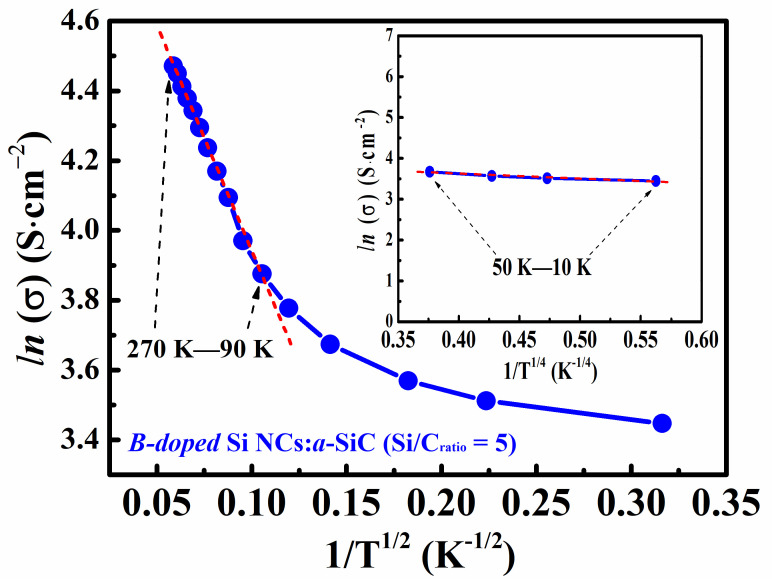
The lnσ plotted as a function of $T^{-1/2}$ of the B-doped Si NCs:*a*-SiC film with *R* = 5; the insert is the lnσ plotted as a function of $T^{-1/4}$ at the temperature from 10 K to 50 K.

**Table 1 nanomaterials-11-02678-t001:** Room temperature Hall measurement.

Sample	Hall Mobility	Carrier Concentration	Dark Conductivity
*B-doped* Si NCs:*a*-SiC film (Si:C_ratio_ = 5)	7.2 cm^2^/V·s	4.6 × 10^19^ cm^−3^	87.5 S∙cm^−1^
*B-doped* Si NCs:*a*-SiC film (Si:C_ratio_ = 2)	4.8 cm^2^/V·s	1.9 × 10^18^ cm^−3^	1.46 S∙cm^−1^
*B-doped* Si NCs:*a*-SiC film (Si:C_ratio_ = 1)	1.7 cm^2^/V·s	8.7 × 10^16^ cm^−3^	0.023 S∙cm^−1^

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
