# Peer review of "Structures, Electronic Properties and Carrier Transport Mechanisms of Si Nano-Crystalline Embedded in the Amorphous SiC Films with Various Si/C Ratios"

_nanomaterials, 2021, doi:10.3390/nano11102678_

Round 1

Reviewer 1 Report

The formation of SiC films containing Si nanocrystals is relevant from the point of view of creating multijunction solar cells based on a monocrystalline silicon substrate. In such structures, the upper cell band gap should be wider than that of the monocrystalline silicon. This condition is realized in SiO2 and Si3N4 films containing Si nanocrystals (SiC-ncSi) (quantum confinement effect). Crystalline SIC with a cubic structure (3C SiC) is also a suitable material for these purposes because of its wide band gap. Compared to SiO2 and Si3N4, SiC has a sufficiently high conductivity and is more suitable for creating solar cells. Currently, a number of methods are used to create SiC-ncSi [1-7]. However, two main problems still remain unresolved: the problem of controlling the nanocrystal size and the problem of defect elimination. Therefore, the results presented in the peer-reviewed article are very actual. They are performed at a fairly good experimental level. However, I have a number of comments both on the presentation of the results and on their discussion.

  1. The abbreviations are usually not used in the abstract.
  2. What is the thickness of the films? Is it the same in all experiments?
  3. The films were deposited on a silicon substrate and a quartz substrate. On which substrates were the results presented in the figures obtained? Do the properties of films deposited on silicon and quartz substrates differ?
  4. In Fig. 1, the obtained optical phonon mode in the nanocrystals is low-frequency shifted, as compared to that in monocrystalline Si. Both the quantum confinement effect and the bond stresses in the nanocrystals can result in the obtained optical phonon properties. These effects are not discussed in any way in the article.
  5. Optical acoustic phonon properties in SiC are not discussed. The intensity of the acoustic phonon mode can provide information about the structural perfection of the films.
  6. In Fig. 2, the scale label is poorly visible.
  7. Table 1. Why does the mobility of charge carriers increases simultaneously with the increase in their concentration?

References

  1. Summonte, C., et al., Silicon nanocrystals in carbide matrix. Solar Energy Materials and Solar Cells, 2014. 128: p. 138-49.
  2. Song, D., et al., Structural characterization of annealed Si(1-x)C(x)/SiC multilayers targeting formation of Si nanocrystals in a SiC matrix. Journal of Applied Physics, 2008. 103: p. 083544.
  3. Ouadfel, M.A., et al., Si-rich a-Si1−xCx thin films by d.c. magnetron co-sputtering of silicon and silicon carbide: Structural and optical properties. Applied Surface Science, 2013. 265: p. 94-100.
  4. Schnabel, M., et al., Self-assembled silicon nanocrystal arrays for photovoltaics. Physica status solidi (a), 2015. 212(8): p. 1649-1661.
  5. Janz, S., P. Löper, and M. Schnabel, Silicon nanocrystals produced by solid phase crystallization of superlattices for photovoltaic applications. Materials Science and Engineering B, 2013. 178(9): p. 542-50.
  6. Shukla, C. Summonte, M. Canino, M. Allegrezza, M. Bellettato, A. Desalvo, D. Nobili, S. Mirabella, N. Sharma, M. Jangir, I. P. Jain. Optical and electrical properties of Si nanocrystals embedded in SiC matrix, Advanced Materials Letters. 2012, 3(4), 297-304.
  7. Weiss, M. Schnabel, S. Prucnal, J. Hofmann, A. Reichert, T. Fehrenbach, W. Skorupa, S. Janz. Formation of Silicon Nanocrystals in Silicon Carbide Using Flash Lamp Annealing, Journal of Applied Physics, 2016. 120, p. 105103.

Reviewer 2 Report

The authors report on the electronic properties and carrier transport mechanisms of Si NCs embedded in amorphous SiC films. For this, samples with three gas ratios [SiH4] / [CH4] are employed to underline the degree of crystallinity by Raman spectroscopy, the size of NCs observed by TEM, beside the optical and electrical properties measured such as the optical bandgap, the hall mobility and the carrier concentration. To further investigate about the carrier transport mechanism, the authors also report about the temperature-dependent conductivity with and without reduced activation energy and underline a specific temperature range where the Mott variable-range hopping, the multiple phonons hopping, and the percolation hopping mechanisms are most likely to occur.

Besides slight improvement in the English part in general, the method used and the presentation of the results are well presented. However, several points need to be clarified as follow:

1)In the experimental part, the substrates used must be more clarified (line 87): what is the resistivity, thickness, and orientation of the p-type silicon wafer? Besides, instead of using “various measurements” the authors must clearly mention when quartz or p-Si wafers are chosen for each specific measurement. After knowing the substrate resistivity, the authors have to mention how they separated the mobility, carrier concentration from the Si NCs/a-SiC from the substrate.

2)In the experiment part, line 112-113: Xc is increased to 90% for R=5, which means that the crystalline part is larger than the amorphous part. However, it may just indicate that the nanocrystals became larger, with a similar concentration for different R. Because the density or concentration of Si-NCs is not mentioned, we can’t conclude that larger Si concentration are really promoting the formation of Si NCs (Lines 112-113).

3)In the figure 2, because the density or concentration may be difficult to obtain, at least a size distribution of Si NCs embedded would be necessary along with the TEM images (using larger image scale than 5nm) for each R condition. A method to extract the size distribution is presented in [1] by using the free software ImageJ. A size distribution would be helpful to confirm whether the formation of Si NCs is promoted or not.

[1]         C.A. Schneider, W.S. Rasband, K.W. Eliceiri, NIH Image to ImageJ: 25 years of image analysis, Nat. Methods. 9 (2012) 671–675. https://doi.org/10.1038/nmeth.2089.

4)In the experimental part, line 133-134: Another possible reason why the bandgaps are larger (2.1-2.2 eV) compared to a-SiC about 1.8 eV is because the NCs have an opening (increasing) bandgap when the size of the NCs decrease and become closer to the Bohr radius [2], based on the Brus equation [3]. For example, thanks to the Brus equation, knowing the optical bandgap and assuming the electron/hole effective mass of the material, one could also estimate the size of the NCs and compared with the size distribution experimentally observed.

[2]         D.C. Hannah, J. Yang, P. Podsiadlo, M.K.Y. Chan, A. Demortière, D.J. Gosztola, V.B. Prakapenka, G.C. Schatz, U. Kortshagen, R.D. Schaller, On the origin of photoluminescence in silicon nanocrystals: pressure-dependent structural and optical studies., Nano Lett. 12 (2012) 4200–5. https://doi.org/10.1021/nl301787g.

[3]         L. Brus, Electronic wave functions in semiconductor clusters: Experiment and theory, J. Phys. Chem. 90 (1986) 2555–2560. https://doi.org/10.1021/j100403a003.

5)Generally, The bandgap in NCs depends on the semiconductor materials, the size of the NCs, and also its surface [4], in the experimental part, line 136: Another possibility of the larger bandgap could be attributed also to crystallized SiC with a bandgap about 3.26 eV … with larger grain, more C may also be incorporated into Si-Si which could be another reason why the bandgap is measured at 2.8 eV. The authors emphasized only the “BGs” (line 137), however this part could be slightly more describe: the meaning of BGs must be mentioned, why does BGs occur? And is it a positive or negative point for your films? Why BGs are most likely to occur over crystallized SiC?

[4]         M. Bürkle, M. Lozac’h, C. McDonald, D. Mariotti, K. Matsubara, V. Švrček, Bandgap Engineering in OH-Functionalized Silicon Nanocrystals: Interplay between Surface Functionalization and Quantum Confinement, Adv. Funct. Mater. 27 (2017) 1701898. https://doi.org/10.1002/adfm.201701898.

6)In Table 1, the hall mobility, and the carrier concentration are well presented. However, these is a lack of comparison with other reference/work, which could be helpful to the reader.

7)Line 201 the Mott-VRH conduction must be described as the Mott variable-range hopping, one or two references would also be appropriate.

8)Line 85: the authors could briefly mention why a dehydrogenation step is important prior to the thermal annealing.

9)Line 261: like the authors mentioned in the discussion part, the hall mobility and carrier concentration are enhanced due to the improvement of crystallinity of the films, but also due to an improved substitutional B doping into Si NCs, which could be added to the conclusion as well.

10)In general the abstract and introduction could be more precise by avoiding colloquial expressions (such as: all sort of, various, different kinds of, etc. …) which will have a more scientific impact on the global manuscript:

In the abstract “all sorts of nanometer devices” is a bit too colloquial, rather name specific devices which already use or will mostly benefit the nanomaterials … like in the introduction part for example by using the nano-electronic and optoelectronic devices.

In the abstract, the following sentence “carrier transport properties of […] incorporation with microstructural characteristics […]” probably need some modifications for clarity, what is incorporated?

In the abstract … “different kind of carrier transport behavior”, again, it is better to clearly name the different transport: such as the Mott-VRH, the MPH, and the percolation hopping model that play an important role in the transport mechanism at specific temperature range.

In the introduction

Line 42: solar cell with a power conversion efficiency reaching about 6.11%.

Line 52: in order to further enhance the performance

Line 55: maybe the values of electron and hole mobilities in a-SiC could be mentioned here.

Lines 56-58: the authors could mention why there is significant challenges and still on-going debate concerning the mechanism of carrier transport in Si NCs.

Line 68: “it is found” could be replace by “Our results underlines” or “We underlines”, for example.

In the introduction the expression “different kinds of conduction mechanism” is a bit large, it could be better to directly name the different mechanisms involved.

In the Experiment part:

“… various measurements” … it is better to directly names the different measurements performed.

In the experiment part line 107: “annealing samples”

In the experiment line 121: the grain size is obviously enhanced with R … (i.e. the grain size increased when R is also increased).

In general, the word “obviously” is often used in the manuscript even if it does not add more pertinent argument in the discussion, so it would be better to just remove it.

Line 183: Avoid colloquial expression such as “as we all know”

After the revision of the mentioned points, I would recommend the present manuscript for publication in Nanomaterial.

Round 2

Reviewer 1 Report

Comments are fully taken into account

Author Response

Thanks for your suggestions. We have checked the manuscript.

Reviewer 2 Report

The authors have answered to each important point mentioned and have carefully revised the present manuscript.

There is only one remaining comment. In the experimental part, the authors forgot to mention the type of substrate used, which is well answered in their reply to reviewers: "a p-type silicon wafer, oriented <100>, 1-3 ohm cm with a thickness of 500 um". I believe that such information is also important to be clarified in the experimental part of the final manuscript.

Therefore, I would recommend the present manuscript for publication in Nanomaterials after the minor revision.

Author Response

Thanks for your suggestions. We have added this information.